# Efficient wastewater sample filtration improves the detection of SARS-CoV-2 variants: An extensive analysis based on sequencing parameters

Angelo Robotto[1], Carlotta Olivero[2]*, Elisa Pozzi[2], Claudia Strumia[2], Camilla Crasà[2], Cristina Fedele[2], Maddalena Derosa[2], Massimo Di Martino[2], Stefania Latino[2], Giada Scorza[2], Andrea Civra[3], David Lembo[3]*, Paola Quaglino[1], Enrico Brizio[1], Denis Polato[2]

1 Environmental Protection Agency of Piedmont (Arpa Piemonte), Torino, Italy, 2 Department of Regional Centre of Molecular Biology, Environmental Protection Agency of Piedmont (Arpa Piemonte), La Loggia, Torino, Italy, 3 Dept. of Clinical and Biological Sciences, University of Turin, Orbassano, Torino, Italy

☯ These authors contributed equally to this work.
* carloliv@arpa.piemonte.it (CO); david.lembo@unito.it (DL)

**Data Availability Statement:** The datasets generated and/or analysed during the current study

## Abstract

During the SARS-CoV-2 pandemic, many countries established wastewater (WW) surveillance to objectively monitor the level of infection within the population. As new variants continue to emerge, it has become clear that WW surveillance is an essential tool for the early detection of variants. The EU Commission published a recommendation suggesting an approach to establish surveillance of SARS-CoV-2 and its variants in WW, besides specifying the methodology for WW concentration and RNA extraction. Therefore, different groups have approached the issue with different strategies, mainly focusing on WW concentration methods, but only a few groups highlighted the importance of prefiltering WW samples and/ or purification of RNA samples. Aiming to obtain high-quality sequencing data allowing variants detection, we compared four experimental conditions generated from the treatment of: i) WW samples by WW filtration and ii) the extracted RNA by DNase treatment, purification and concentration of the extracted RNA. To evaluate the best condition, the results were assessed by focusing on several sequencing parameters, as the outcome of SARS-CoV-2 sequencing from WW is crucial for variant detection. Overall, the best sequencing result was obtained by filtering the WW sample. Moreover, the present study provides an overview of some sequencing parameters to consider when optimizing a method for monitoring SARS-CoV-2 variants from WW samples, which can also be applied to any sample preparation methodology.

are available in the ENA repository, Accession number PRJEB56331.

**Funding:** Arpa Piemonte received no specific funding for this work. Professor DL was supported by EU funding within the MUR PNRR Extended Partnership initiative on Emerging Infectious Diseases (Project no. PE00000007, INF-ACT). The funders had no role in study design, data collection and analysis, decision to publish, or preparation of the manuscript.

**Competing interests:** The authors have declared that no competing interests exist.

## 1. Introduction

In the midst of the Severe Acute Respiratory Syndrome Coronavirus - 2 (SARS-CoV-2) pandemic outbreak, Wastewater Based Epidemiology (WBE) was proposed as an epidemiological tool approach for disease surveillance [1–3]. Several studies have described that SARS-CoV-2 replication was localized in the upper and lower respiratory tract, but the virus has also been reported to colonize the gut and hence shed via human stool and therefore found in untreated wastewater (WW) [4, 5]. Furthermore, it has been demonstrated that the viral load among asymptomatic and symptomatic infections was comparable [6], indicating that the silent disease transmission among asymptomatic hosts has also been the cause of the persistent outbreaks, despite most symptomatic cases being immediately isolated following protective measures [7, 8].

To observe the spread and persistence of the SARS-CoV-2 virus in the population, different countries have implemented the monitoring of SARS-CoV-2 in sewage and this has been proven to be a useful additional predictive tool for the COVID-19 pandemic [9–13].

On the 17th of March 2021, the European Commission (EU) published Recommendation 2021/472, announcing a common approach to establish systematic surveillance of SARS-CoV-2 and its variants in WW in the EU. The Recommendation stated that WW monitoring should be considered an independent and complementary approach to COVID-19 surveillance and testing strategies.

In the past years, Arpa Piemonte, the Regional Environmental Protection Agency of Piedmont (Italy), has been involved in the collection and analysis of the influent untreated WW of the main Wastewater Treatment Plants (WWTPs) in Piedmont, to detect the presence of SARS-CoV-2 virus in that matrix. Data published by our group [14] disclosed that during the second and third pandemic waves, from October 2020 to March 2021, the trend of virus in sewers foreshadows the trend of positive nasopharyngeal swabs, suggesting that the SARS-CoV-2 monitoring in the WW matrix could be considered a complementary approach to the surveillance of the infection level in the population. Moreover, besides monitoring the presence of SARS-CoV-2 in WW, this surveillance aims to detect the variants of the virus, which, being an RNA virus, mutates with a high frequency [15].

Whole-genome sequencing (WGS) of viruses has become a powerful tool for studying emerging infectious diseases while monitoring public health and amplicon sequencing for viral WGS is the preferable approach for library preparation since it is simple, cost-effective and sensitive for the detection of genomes from low viral load samples [16, 17]. However, most of the viral genome sequences used to monitor the evolution of viruses have been usually obtained directly from clinical samples, as described in studies focused on Ebola and Zika viruses, influenza viruses, and DNA viruses, like human cytomegalovirus (HCMV), but not much monitoring has been performed using sequences obtained from WW samples [17, 18].

With the focus on combining viral WGS and genomic WBE, it is important to emphasize that the complexity of the WW matrix is challenging for pathogen detection: in particular, the integrity of viruses could be affected by temperature, pH, solids, micropollutants such as humic acids, and the presence of other microorganisms [19, 20]. Earlier than the development of WBE for the COVID-19 pandemic, previous studies showed that the detection of viral pathogens in sewage depended on the concentration and extraction methods used [21–23]. With the rise of the COVID-19 pandemic, the absence of guidelines for SARS-CoV-2 recovery from WW induced several groups to test different procedures of sampling, samples storage, concentration, extraction and viral detection, most of them focused on recovering the highest quantity of virus for RT-qPCR analysis [24–26]. Only recently some works [23, 27–32] have shifted this focus to improve genomic coverage for a good sequencing outcome.

In this study, we aimed to compare the treatment of the WW samples and/or the extracted RNA to elucidate how they affect the sequencing outcomes. We analyzed three samples from three different WWTPs and compared the results by means of several sequencing parameters, with the goal of obtaining high-quality sequencing data as the basis for reliable monitoring of SARS-CoV-2 variants present in WW.

## 2. Materials and methods

### 2.1 WW samples collection

The untreated WW was collected from three different WW treatment plants (WWTPs) in the Piedmont region, located in North-West Italy: Castiglione Torinese, Alessandria and Cuneo. From 500 mL to 1000 mL of 24-hour flow composite samples were collected using automatic sampling devices. Samples were kept at 4˚C during transport and storage. All samples were processed within 24 hours after sampling. The samples were collected within the institutional activities of Arpa Piemonte.

### 2.2 Filtration

Before the filtration step, 40mL of the composite WW sample selected for filtration were centrifuged at 3000 x g for 10 minutes to remove solids. The initial volume of 40 mL was chosen according to the protocol of the Wizard® Enviro TNA kit protocol (Promega, Milan, Italy). The supernatant was then filtered through a 0.45 μm polyethersulfone (PES) filter followed by a 0.2 μm PES filter. At this point, Total Nucleic Acid (TNA) isolation was performed on both filtered and not filtered samples. All WW samples were handled in a Biosafety Level 2 laboratory under a biological cabinet.

### 2.3 Extraction of Total Nucleic Acid from WW samples

The filtered and not filtered samples were incubated with 0.5mL Protease Solution for 30 minutes at room temperature (RT). The non-filtered samples were centrifuged at 3000 x g for 10 minutes to remove solids.

TNA was extracted using the Wizard® Enviro TNA kit (Promega, Milan, Italy) according to the manufacter's instructions. TNA was eluted in 40 μL of RNase-free water and stored at -80˚C.

### 2.4 TNA treatment: DNase I digestion, purification and concentration of extracted TNA

35 μL of the samples selected for the DNase treatment, purification and concentration phases were treated with 1.5 U of RQ1 RNase-free DNase (Promega, Milan, Italy) for 30 minutes at 37˚C in a final reaction volume of 45 μL according to the manufacturer's protocol.

Samples were then purified using the ReliaPrep™ RNA Clean-Up and Concentration System (Promega, Milan, Italy) according to the manufacturer's technical manual. RNA was eluted with 18 μL of nuclease-free water.

### 2.5 RT-qPCR and quantification of viral load in WW samples

5 μL of TNA or purified RNA from each extracted sample were used to perform one-step RT-qPCR using the GoTaq® Enviro Wastewater SARS-CoV-2 System (Promega, Milan, Italy), hydrolysis probe-based reverse transcription kit, which amplifies the Nucleocapsid (N1) region of the SARS-CoV-2 genome. A standard curve, ranging from $10^3$ copies (cp)/μL to 0.5 cp/μL, was generated using a standard genomic RNA from Severe Acute Respiratory Syndrome-related Coronavirus 2 (ATCC® VR-1986D™) (LGC Standard, UK). Standard curves

were generated by serial 10-fold dilution with nuclease-free water. A no-template control (NTC) was analyzed alongside the experimental samples, using nuclease-free water. Samples and standard curve points were both analyzed in duplicate using 5μL of TNA/RNA for each WW sample and standard point. The final reaction volume was 20 μL.

## 2.6 SARS-CoV-2 amplicon library preparation, quality control and sequencing

NGS library preparation of the whole viral genome was performed using the QIAseq® DIRECT SARS-CoV-2 protocol (until April 2021 –Qiagen, Milan, Italy) with minor modifications for WW samples. For reverse transcription, 13 μL of RNA, 1 μL of RP primer diluted 11-fold, 4 μL of Multimodal RT Buffer 5x and 1 μL of RNase inhibitor were mixed and incubated at 25°C for 10 minutes followed by two incubations, at 42°C for 30 minutes and 85°C for 5 minutes respectively, before holding at 4°C. For cDNA amplification, we used multiplexed primer pools in a high-fidelity multiplex polymerase chain reaction (PCR) to generate approximately 225–275 base pair (bp) amplicons. A multiplex-PCR was performed for each of both SARS-CoV-2 primer pools (DIRECT SARS-CoV-2 Pool #1 or Pool #2) by combining 12.5 μL of QIAseq 2X HiFi Master Mix, 8 μL of nuclease-free water and 2.5 μL of RT cDNA. The mix was incubated at 98°C for 2 minutes followed by 35 cycles at 98°C for 20 seconds and 63°C for 5 minutes before holding at 4°C. The two enriched pools per sample were then pooled into a single tube and purified using a QIAseq Bead Cleanup (Qiagen, Milan, Italy) according to the manufacturer's instructions. Samples were eluted in 30 μL of nuclease-free water and quantified using a Qubit High Sensitivity Kit (ThermoFisher, Milan, Italy) according to the manufacturer's instructions to normalize each sample to a concentration of approximately 100 ng/ μL. Following the quantification and normalization phases, SARS-CoV-2 enriched samples were amplified and indexed using a high-fidelity amplification reaction: 23 μL of each enriched sample were mixed with 25 μL of QIAseq 2X HiFi Master Mix and 2 μL of specific dual-Index from the QIAseq DIRECT UDI index plate. The mix was incubated at 98°C for 2 minutes followed by 10 cycles at 98°C for 20 seconds, 60°C for 30 seconds and 72°C for 30 seconds before holding at 4°C. The final amplicon library from each sample was purified using a QIAseq Bead Cleanup (Qiagen, Milan, Italy) and eluted in 25μL nuclease-free water following the manufacturer's instructions. Library quality was assessed using the 2100 Bioanalyzer System (Agilent Technologies, Milan, Italy) and quantified using the Qubit HS dsDNA assay (ThermoFisher, Milan, Italy). 5 μL of the library from each sample were combined and thoroughly mixed to generate a single pool. The pool was diluted according to Illumina's standard protocol. The Illumina MiSeq instrument was used to generate dual 10 bp indexes and 149 bp paired- end (PE) length reads. A Miseq V2 Micro cartridge (Illumina, Milan, Italy) was used and one million PE reads was the aim for sequencing depth for each sample.

## 2.7 Statistical analysis

Experiments were performed on three biological replicates (n value = 3). Values are presented as mean and standard deviation, followed by the ± symbol. The normal distribution of the data was assessed by visual inspection analyses (histogram and Quantile-Quantile (QQ) plot) and statistical analysis (Shapiro-Wilk test). The significance of the data was calculated using two-way ANOVA (Analysis of variance) and paired Student's t-test. Statistical calculations were performed using GraphPad Statistics Software Version 6.0 (GraphPad Software, Inc., USA) and Rstudio version 2023.09.0+463 with R version 4.3.1. p-values of $< 0.05$ were considered statistically significant.

## 2.8 Bioinformatic analysis

Illumina MiSeq instrument was used to obtain demultiplexed fastq data. For sequencing quality, FastQC v0.11.9 was used (https://www.bioinformatics.babraham.ac.uk/projects/fastqc/). The bioinformatics pipeline to trim adapters and primers, generate the consensus sequences and evaluate the genomic mutations was performed using the CLC Genomics Workbench platform 22.0 (Qiagen, Milan, Italy). The SARS-CoV-2 reference sequence used for the alignment was MN908947.3. As configurable parameters, the minimum frequency of nucleotide variation and the minimum average base quality were set at 3% and 25.0 respectively. In addition, a minimum coverage of 30X was considered for each position.

The Dragen metagenomics pipeline 3.5.11 (https://emea.illumina.com/products/by-type/informatics-products/basespace-sequence-hub/apps/dragen-metagenomics-pipeline.html) was used to perform taxonomic classification of the reads using the Extended Kraken2 algorithm (March 2020) [33].

The list of mutations belonging to the Omicron spike gene of the BA.1 variant was downloaded from the constellations repository of cov-lineage (https://github.com/covlineages/constellations/blob/feb3b11fcb4f7d39bfa9dd1d38852932561e6295/constellations/definitions/cBA.1.json#L25).

# 3. Results and discussion

## 3.1 Samples preparation

With the purpose of analysing whether the filtration of WW samples and/or the TNA purification (consisting of DNase I digestion, purification and concentration of the extracted TNA) could improve the sequencing result, three WW samples from different WWTPs were processed in different ways. In accordance with the filtration and treatment steps, the following four experimental conditions were generated: F-NT: WW filtration - No TNA treatment; F-T: WW filtration - TNA treatment; NF-NT: No WW filtration - No TNA treatment; NF-T: No WW filtration - TNA treatment (S1 Table).

Recently, many papers have been published focusing on methods to concentrate SARS-CoV-2 from WW and subsequently extract its RNA for RT-qPCR and sequencing [34, 35]. After an extensive analysis of the different methodologies, we chose a direct capture method that concentrates the virus and simultaneously extracts the TNA [25]. This choice was based on the following strengths: i) speed and simplicity of the method; ii) no requirement for an ultracentrifugation step; iii) low WW input volume (less than 50 mL) and iv) it has been demonstrated to be reliable and superior in comparison to other methods [25, 34, 36]. Some papers have also highlighted the importance of pre-filtration of WW to remove bacterial cells and debris [30, 34, 37–39] as well as the post-extraction purification of RNA, a treatment that removes inhibitors commonly found in WW samples such as humic acids [25, 39].

## 3.2 RT-qPCR and viral load

Prior to library preparation, RT-qPCR was performed on WW samples with the purpose of evaluating the viral load of each sample. The aim of this approach is to assess which experimental method allows the highest copy number of SARS-CoV-2 to be obtained and used as input for the library preparation in order to generate enough yield and reliable output of the sequencing result.

Ct values of the SARS-CoV-2 N1 gene from each sample were interpolated to a standard curve. Among the treated samples, the NF-T samples, which were not filtered but only treated, showed the highest concentration of SARS-CoV-2 (average concentration of the

three samples 71.27±20.75 cp/μL), followed by the NF-NT samples (average of the three samples 56.04±26.63 cp/μL). In addition, the filtered conditions (both NT and T) showed an average concentration of three samples of 46.38±19.28 cp/μL and 56.91±16.79 cp/μL of SARS-CoV-2 respectively (S2 Table).

Since 5 μL is the suggested input volume for library preparation, mean viral copies input per each condition were shown in Fig 1. It was observed that the concentration of SARS-CoV-2 was highest in the NF-T samples (average concentration of the three samples 356.36 ± 103.73 copies input), followed by the NF-NT samples (average of the three samples 280.19±108.73 copies input). The filtered and treated samples (F-T) showed an average of 284.56±83.97 copies input, whereas the filtered but not treated samples (F-NT) showed an average of 231.92 ±96.40 copies input of SARS-CoV-2 (Fig 1). These results suggest that the copies number obtained from the F-NT condition were significantly lower compared to the NF conditions (both NT and T).

These results are in line with other work reporting that one reason for lower viral load detection of SARS-CoV-2 in WW samples that were prefiltered prior to RNA extraction could be due to the elimination of solid particles, which, especially for viruses highly associated with solids, such as SARS-CoV-2 in WW, could carry a significant amount of RNA [40, 41]. On the contrary, other studies [42–44] reported that the removal of debris by means of filtration through a 0.22 μm PES membrane was the best method for pretreatment of WW samples prior to SARS-CoV-2 RNA extraction with the purpose of removing bacterial cells and debris. Moreover, the number of viral cp input into library preparation was not identical within or between conditions, in order to assess whether more viral cp could result in an increased sequencing yield. This point is crucial in the view of applying and standardizing this methodology for high throughput SARS-CoV-2 monitoring from WW, since the library volume input is the same, regardless of the viral titer.

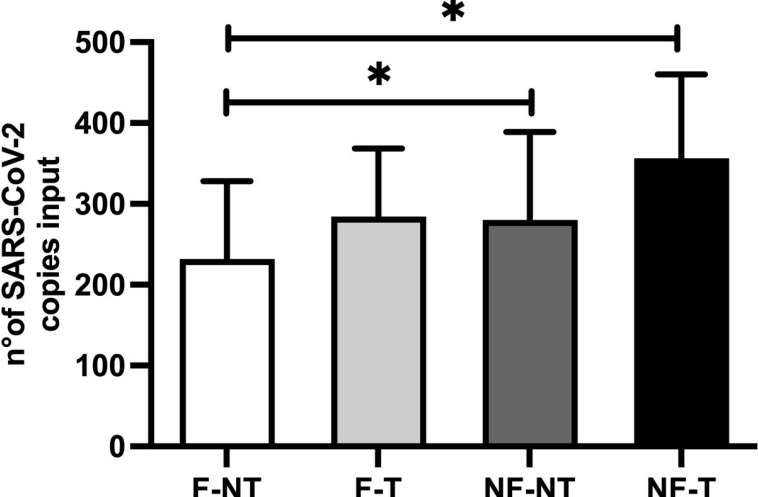

**Fig 1. Effect of filtration (F) and purification treatment (T) on the number of SARS-CoV-2 viral copies used as input for the library preparation.** Copy number yield for the N1 gene was obtained by RT-qPCR from three biological samples processed according to four experimental conditions: F-NT: WW filtration - no RNA treatment; F-T: WW filtration - RNA treatment; NF-NT: no WW filtration - no RNA treatment; NF-T: no WW filtration - RNA treatment. Two-way ANOVA (p ≤ 0.05). Bar plots show mean and standard deviation and asterisks show significance level upon t-test (not significant (no asterisks) = p > 0.05; * = p ≤ 0.05; ** = p ≤ 0.01; *** = p ≤ 0.001; **** = p ≤ 0.0001, n = 3).

### 3.3 Library QC

To assess the quality and check the size distribution of the library fragments, samples were loaded onto an HS DNA Assay Chip at the same concentration and run on the 2100 Bioanalyzer. The concentration was kept identical in all samples to assess whether the different conditions and input concentrations could affect the amplification of the fragments. S3 Table summarizes the average bp size of the fragments.

The size of the fragments and the intensity of the bands vary slightly between the different conditions. The main band is visible in all the samples indicating that fragments of the same size were generated in each sample (Fig 2). The library band corresponds to approximately 75 seconds (350–400 bp). Furthermore, smaller and lighter bands around 60 seconds were visible in the samples from the NF-NT and NF-T conditions, hinting that short and unspecific fragments were generated in these samples, regardless of the amount of RNA input quantity, but only dependent on the condition (Fig 2A). Moreover, these bands are also visible as peaks in the electropherograms associated with the gel: the higher peaks around 60 seconds are from the NF-NT and NF-T samples for all the three WWTPs, while the lower peaks are from the F-NT and F-T samples (Fig 2B).

We speculate that the filtration step removes some unspecific and off-target nucleic acid sequences that might otherwise be amplified in the library preparation. In addition, Trujillo et al. argue that the removal of solids by filtration facilitates the downstream processing steps by eliminating genomic contamination, such as nucleic acid sequences from other species, that could be unspecifically amplified and might hinder the ability to deeply sequence SARS-CoV-2 [37]. Therefore, from a sequencing perspective, the filtration of WW samples improves the specificity of amplification by removing the potential off-target nucleic acid sequences that are present in such a complex matrix as also demonstrated in previous studies [29, 30].

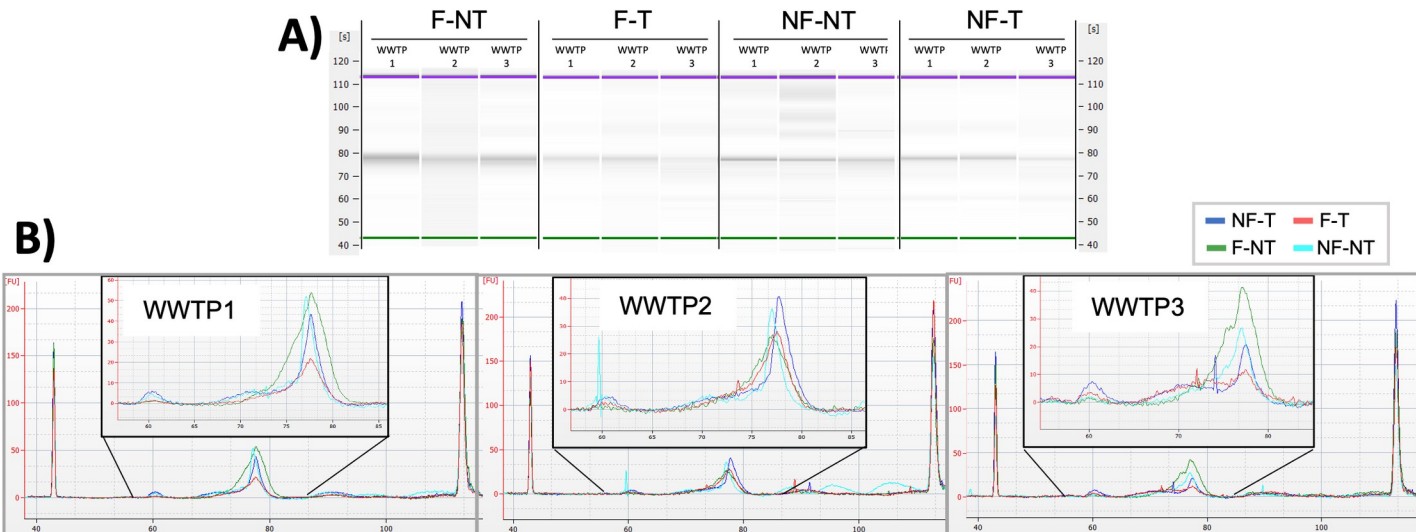

**Fig 2. Library quality assessment using the Agilent 2100 Bioanalyzer System-High Sensitivity DNA Kit.** A) Data displayed as a gel-like image. The upper (purple) and the lower (green) bands represent the internal markers. B) Electropherogram traces represent the expected size distribution of the library fragments at 75 seconds (350–400 bp). The color of the line indicates the different conditions: F-NT (green), F-T (red), NF-NT (light blue) and NF-T (blue). The left (approximately 45 seconds) and far-right peaks (approximately 115 seconds) are internal markers.

### 3.4 Amplicon sequencing of SARS-CoV-2 RNA from WW samples and evaluation of Quality Control (QC) metrics

**3.4.1 QC: Raw reads data.** To blindly assess the origin of the reads and whether any off-target nucleic acid fragments were sequenced, the sample's fastq files were uploaded to the Illumina DRAGEN Metagenomics pipeline. The pipeline performs a metagenomics classification workflow capable of detecting and quantifying microbial and viral pathogen sequences with high analytical specificity and sensitivity. The percentage of the means of the raw reads classified as belonging to SARS-CoV-2 is shown in Fig 3A. In the F-NT samples, 62% ± 7 of the reads were classified as SARS-CoV-2; whereas, in the F-T, NF-NT and NF-T samples, only ≤40% of the reads were classified as aligned to SARS-CoV-2. Furthermore, as shown in Fig 3B, the proportion of reads classified as belonging to bacteria was significantly reduced only in the F-NT samples (average of the three samples: 19.67% ± 1.7 for F-NT) compared to the not filtered samples (average of the three samples: 33.67% ± 2.9 for NF-NT and 37.33% ± 4.6 for NF-T).

Interestingly, the F-NT is significantly different compared to the NF conditions (NF-NT and NF-T) for both the % of raw reads classified as belonging to SARS-CoV-2 and as belonging to bacteria. This result reinforced the point discussed in paragraph 3.3, suggesting that the filtration step of the WW sample may indeed remove the source of bacteria.

Moreover, the raw reads were analyzed for the quality score across all bases using FastQC software to understand whether the data unveiled any issues that might affect further analysis. Remarkably, only the fastq files from the F-NT condition for the three WWTPs were classified as "passed", indicating that the median quality score for any base is greater than 25 (S1 Fig and S4 Table). This means that in terms of the quality of the sequencing output, the filtered samples had the highest quality score across all bases, regardless of the treatment. This may be due to the fact that the WW filtration removes off-target sequences that could be randomly amplified during the library process and generate unspecific and short sequences, as also seen in Fig 2. Specifically for Illumina sequencing (Sequencing By Synthesis technology), these unspecific and short reads could negatively affect flow cell saturation, clustering formation and base call quality score as, especially for short sequences, if the sequencing read length is longer than the library insert size, sequencing may continue through the Rd1SP-Rd2SP (Read 1 and 2 Sequencing Primer), index and adapter sequences, resulting in a decrease in quality score.

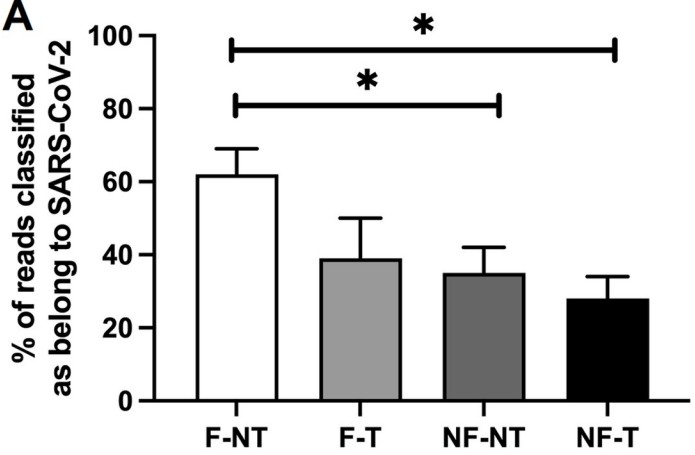 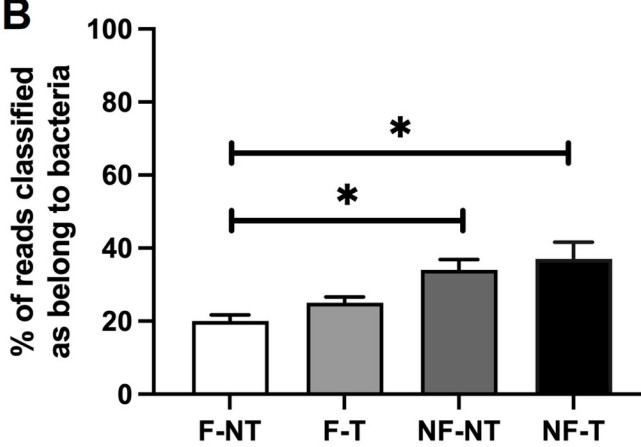

**Fig 3. Taxonomic classification of reads using the Dragen metagenomics pipeline.** A) Percentage of reads classified as belonging to SARS-CoV-2 (Two-way ANOVA ($p \leq 0.05$)); B) Percentage of reads classified as belonging to Bacteria (Two-way ANOVA ($p \leq 0.01$)); Bar plots show mean and standard deviation and asterisks show significance level upon t-test (not significant (no asterisks) = $p > 0.05$; * = $p \leq 0.05$; ** = $p \leq 0.01$; *** = $p \leq 0.001$; **** = $p \leq 0.0001$, n = 3).

**3.4.2 QC: Output of the post-trimmed alignment to the SARS-CoV-2 genome.** To investigate whether the four different sample preparation conditions affected the sequencing output, some selected sequencing parameters were extensively analyzed following the post-trimmed alignment to the SARS-CoV-2 genome. The four groups were compared independently and 1 million reads in Paired-End (PE) and a read length of more than 100 bp were assigned to each sample, as recommended by the EU Recommendation 2021/472.

*3.4.2.1 Read mapping.* We assessed the percentage of reads that were mapped and not mapped to the SARS-CoV-2 reference genome out of the total reads produced. Numerous mapping tools are available, such as Bowtie2, GEM3, BWA-MEM, SOAP2, Novoalign and CLC Genomics Workbench (Qiagen) and they are widely used in re-sequencing studies [45, 46]. In this study, CLC Genomics Workbench was used as the mapping tool. In the samples analyzed, many sequences mapped to SARS-CoV-2 genomes, although with some differences in quantity. The F-NT condition showed a higher percentage of mapped reads (95.96% ± 1.00) compared to the F-T (85.11% ± 6.06), NF-NT (90.97 ± 0.98) and NF-T (85.13% ± 6.64) conditions, although the difference is not significant (S2 Fig). Conversely, the F-NT condition had the lower percentage of not-mapped reads (F-NT samples 4.04% ± 1.00; F-T 14.89% ± 6.06; NF-NT samples 9.03 ± 0.98; NF-T 14.87% ± 6.64) (data not shown). The percentage of reads assigned to SARS-CoV-2 was much higher for all conditions using the CLC genomic workbench software than the Kraken pipeline, possibly because the raw reads used in the Kraken classification include many short reads that are filtered out before direct mapping using the CLC genomic workbench.

*3.4.2.2 SARS-CoV-2 genome coverage ≥ 30X.* Using the CLC Genomic Workbench platform, we evaluated the percentage of target region positions with genome coverage ≥ 30X. This value refers to the percentage of the SARS-CoV-2 genome covered at least 30 times by the sequencing output reads.

As expected, the output varied among the four approaches. Fig 4 shows that the average percentage of target region positions with genome coverage ≥ 30X was above 92% for both

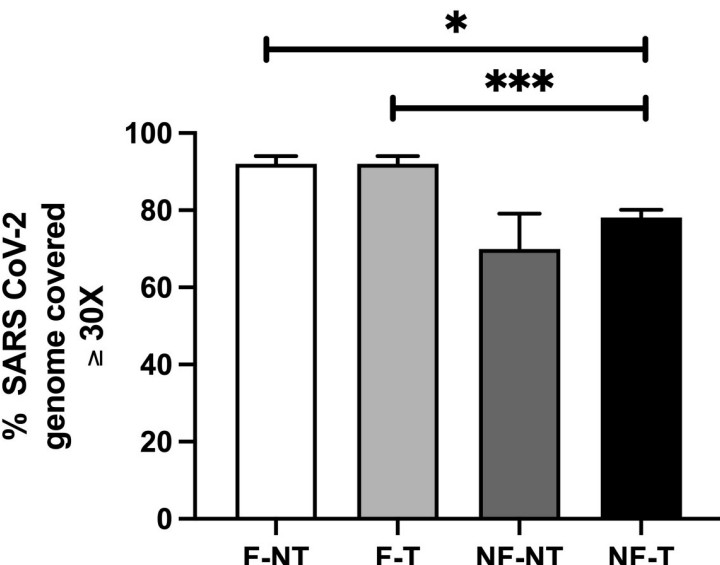

**Fig 4. Percentage of the SARS-CoV-2 genome covered at least 30X.** Percentage of the SARS-CoV-2 genome which is covered at least 30 times by the sequencing output reads. Each condition is the mean of three biological replicates (Two-way ANOVA (p ≤ 0.05)). Bar plots represent mean and standard deviation, and asterisks indicate significance level upon t-test (no asterisks = p > 0.05; * = p ≤ 0.05; ** = p ≤ 0.01; *** = p ≤ 0.001; **** = p ≤ 0.0001, n = 3).

F-NT and F-T samples (average of the three samples 92% ± 2.2 for F-NT and 92% ± 2.1 for F-T) compared to NF-NT and NF-T samples (average of the three samples of 71% ± 11.0 for NF-NT and 78% ± 2.0 for NF-T).

Notably, the library input viral load (Fig 1) did not correlate with the % of SARS-CoV-2 genome coverage ≥ 30X: the F-NT condition had lower average library cp input compared to NF-T, however, the % of SARS-CoV-2 genome coverage ≥ 30X of the F conditions was significantly higher compared to the NF-T condition. Overall, the higher % of SARS-CoV-2 genome coverage ≥ 30X was significantly detected in filtered samples (treated or not) compared to the not filtered, only treated samples (NF-T). In fact, the RNA purification treatment did not significantly increase or decrease the SARS-CoV-2 genome coverage ≥ 30X on either filtered or not-filtered samples.

### 3.5 Identification of low-frequency mutations

Since the WW sample contains SARS-CoV-2 sequences from many different individuals (symptomatic and asymptomatic), there is a need to detect not only the consensus genome sequence generated from the highest frequency mutations, but also mutations that occur at lower frequencies [47].

In order to investigate in detail which of the four experimental conditions allows the detection of low-frequency mutations and thus tracking SARS-CoV-2 variants, a comprehensive analysis was performed at the level of individual mutations resulting from the sequencing of SARS-CoV-2 for each sample. First, to avoid false results, three standard criteria were selected on the CLC Genomic Workbench software: a minimum frequency of nucleotide variant of ≥3%, a minimum average base quality of 25, and a coverage of ≥30X. In addition, to detect mutations at a very low frequency, filters of ≤50%, ≤10%, or ≤5% mutation frequency were selected while maintaining the coverage and minimum average base quality as described above.

The initial number of nt and AA mutations was counted and normalized to the total number of mutations counted for each respective sample. For both nt and AA mutations, there was no significant difference when the mutation frequency of ≤50% was selected (Fig 5A and 5B respectively), whereas, for both ≤10% and ≤5% nt and AA mutation frequencies, the filter

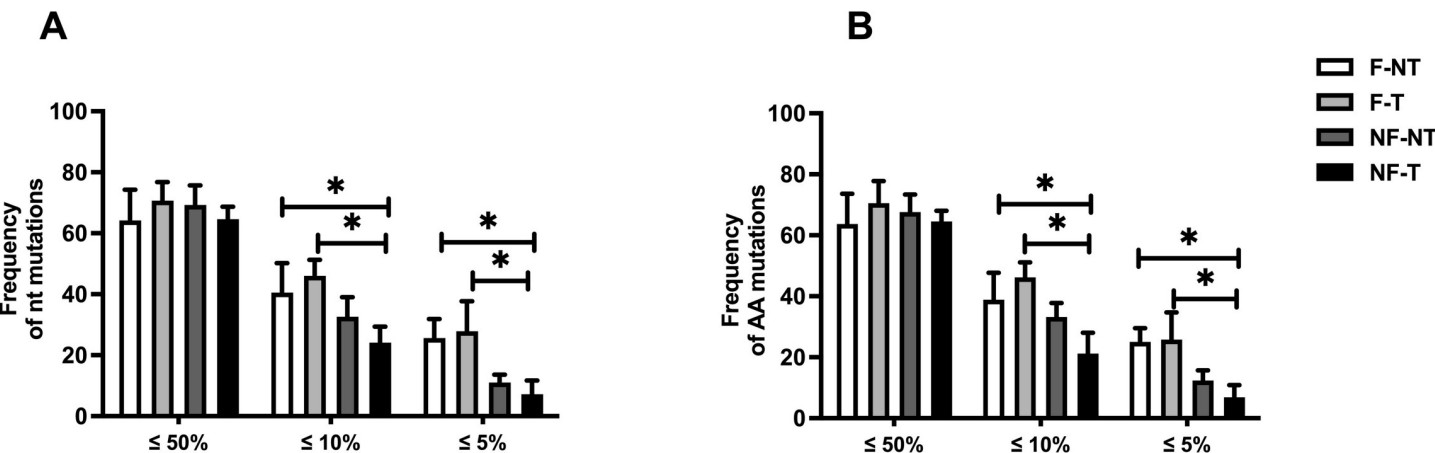

**Fig 5. Frequency of nt and AA mutations with a mutation frequency of ≤ 50%, ≤ 10% or ≤ 5% and genome coverage ≥30X.** A) Frequency of nucleotide mutations with a cut-off mutation frequency of ≤ 50%, ≤ 10% or ≤ 5% and genome coverage ≥30X (Two-way ANOVA (p ≤ 0.001)). B) Frequency of amino acid mutations with a cut-off mutation frequency of ≤ 50%, ≤ 10% or ≤ 5% and coverage ≥30X (Two-way ANOVA (p ≤ 0.001)). Bar plots show mean and standard deviation and asterisks show significance level upon t-test (no asterisks = p > 0.05; * = p ≤ 0.05; ** = p ≤ 0.01; *** = p ≤ 0.001; **** = p ≤ 0.0001, n = 3).

conditions (F-NT and F-T) were significantly different compared to the NF-T condition (Fig 5A and 5B respectively), showing a higher mutation frequency. In addition, the filter conditions (F-NT and F-T) showed a higher trend in mutation frequency compared to NF-NT, although the difference is not significant. These results suggested that the WW filtration step increases the detection of very low-frequency mutations, allowing for the monitoring of emerging variants.

In order to investigate in depth whether the sequencing of the samples treated under different conditions could generate comparable sequences and identify the same mutations belonging to a variant, we compared the sequencing output with the list of mutations characterizing the spike gene of the Variant of Concern (VOC) Omicron BA.1, the variant that was predominantly circulating at that time of the year in the area served by the WWTPs analyzed (Fig 6).

The heatmap showed the percentage of frequency for each mutation colored from yellow (3% of frequency) to red (100% of frequency). The black dots were associated with the mutations that did not satisfy the cut-off of 30X of genome coverage. The BA.1 mutations TACCATG21765-, A67V and G496S did not meet the 30X genome coverage threshold. These results showed that for the NF-NT and NF-T conditions, the number of mutations identified as matching the BA.1 spike mutation list was lower than the corresponding sample processed according to the F-NT or F-T conditions (F-NT: 9±0.82; F-T: 9.33±0.47; NF-NT: 4.67±2.36; NF-T: 5±0.82). Furthermore, the trend of mutation frequency was higher in the F conditions (both T and NT) compared to the NF conditions (both T and NT) for each WWTP compared (Fig 6).

Fig 7 provides a comprehensive summary of the results obtained from the experiments, listing some of the sequencing parameters considered in this work. They are divided into two

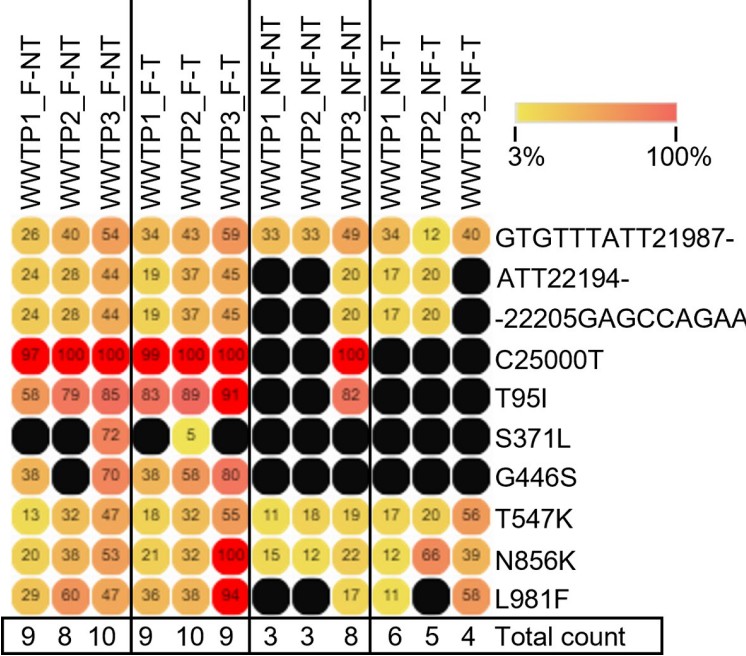

**Fig 6. Omicron BA.1 nt and AA spike gene mutations.** The heatmap shows the presence of spike gene mutations associated with the Omicron BA.1 variant in all the experimental samples analyzed. Each row represents one mutation. The number inside the bullet represents the mutation frequency. The total number of mutations detected is shown at the bottom of each column. Black circles represent mutations with less than 30X coverage. The heatmap was generated with Morpheus, https://software.broadinstitute.org/morpheus.

**Fig 7. Summary heatmap of the evaluated post-sequencing parameters.** The summary heatmap shows the results obtained from the analysis of the different sequencing parameters for each experimental condition (top right columns: F-NT, F-T, NF-NT, NF-T). Dark blue color represents 0%, red color represents 100% while the white color represents 50%. The color scale is relative to the highest and lowest value for each parameter among the treatments.

main categories: the first one examines the parameters related to the raw fastq files and the second one concerns the parameters related to the alignment to the SARS-CoV-2 genome. Each category is further subdivided into subsections and the results obtained from each experimental condition analyzed were associated. Overall, the results for the NF-NT and NF-T conditions were similar and worse compared to the F-NT and F-T conditions for all parameters analysed, except for the SARS-CoV-2 copies number input which was statistically significantly higher in the NF-NT and NF-T conditions compared to F-NT condition (Fig 1). On the contrary, the results obtained from the F-NT and F-T conditions were similar: 2 out of 8 conditions were better in the F-NT (% of reads classified as Bacteria, % of reads classified as SARS-CoV-2); 2 out of 8 in the F-T ($\leq$10% nt, $\leq$10% AA), the remaining 4 out of 8 parameters (% of target region with coverage $\geq$ 30X, $\leq$5% nt, $\leq$5% AA, No. of nt and AA mutations associated with Omicron BA.1 Spike gene) were comparable between the two experimental conditions. Moreover, the fastq Quality Control assay resulted in "passed" for the F-NT condition, whereas "warning" and "failed" were associated with the F-T, NF-NT and NF-T conditions respectively (S4 Table).

Overall, although the RNA treatment increased the percentage of reads classified as belonging to bacteria and decreased the percentage of raw reads classified as belonging to SARS-CoV-2, for the two aforementioned conditions (F-NT and F-T) the percentage of SARS-CoV-2 genome coverage $\geq$30X was similar, as well as the percentage of low-frequency mutations ($\leq$10% and $\leq$5%) and the number of mutations associated with the Omicron BA.1 variant. Thus, these observations support the idea that the filtration of WW samples is relevant to obtain a high-quality SARS-CoV-2 sequencing result and the treatment of the RNA upon WW

filtration could be considered to increase the yield when applied to other RNA extraction methods and/or different library preparation.

Ultimately, it is important to acknowledge the limitations of this work and the need for future work. First, only a limited number of biological samples were compared for the four experimental conditions. Second, the range of Ct for the WW samples fluctuated from 30 to 32, consistent with the peak of positive swab tests for SARS-CoV-2 reported during the corresponding period. We acknowledge that typically the Ct values of WW samples tend to be higher and hence complicate the achievement of high-quality sequencing outcomes. Therefore, more biological replicates under challenging conditions (Ct>35) could strengthen the conclusions obtained from our validation experiments and should be evaluated in future studies. Although this data set is limited in terms of sample number and low Ct values, it shows that there is a potential to obtain a good sequencing result by applying the filtration of the WW samples prior to extraction and, depending on the scenario, in combination with the DNase treatment of the extracted TNA.

## 4. Conclusions

Since the publication of Recommendation 2021/472 by the EU on the 17[th] of March, suggesting a common approach to establish systematic surveillance of the SARS-CoV-2 and its variant in wastewater in the EU, many research groups involved in environmental pathogen surveillance have developed different methodologies to monitor the presence of SARS-CoV-2.

Here, we compared different treatments of wastewater and/or extracted TNA, as pretreatments before the library preparation, to obtain a good-quality sequencing outcome. To choose the best treatment, the results were analyzed based on the output of some selected sequencing parameters. The best sequencing result was obtained by filtering the WW samples.

In conclusion, it is crucial not only to focus on viral load recovery, but also to evaluate the sequencing quality parameters in order to select the most appropriate method for monitoring newly emerging mutations. In addition, the identification of low-frequency mutations is very important for the tracking of SARS-CoV-2 variants circulating in the population and thus for the control of the COVID-19 pandemic.

## Supporting information

**S1 Fig. Per-base sequence quality plots.** Warning: a warning is issued if the lower quartile for any base is less than 10, or if the median for any base is less than 25. Failure: This module raises a failure if the lower quartile for any base is less than 5 or if the median for any base is less than 20.
(PDF)

**S2 Fig. Percentage of reads mapped to the SARS-CoV-2 genome.** Mapped reads refer to those reads that align directly to regions on the SARS-CoV-2 reference genome. Bar plots represent the mean and standard deviation.
(PDF)

**S1 Table. Scheme of the four experimental conditions.**
(PDF)

**S2 Table. Ct values and cp/µL of SARS-CoV-2 N1 gene from each sample.**
(PDF)

**S3 Table. The average bp size distribution of the library fragments run on a Bioanalyzer 2100.**
(PDF)

**S4 Table. Comparison of the quality score across all bases of each raw fastq file using FastQC software.** A Warning is issued if the lower quartile for any base is less than 10, or if the median for any base is less than 25. A Failure is reported if the lower quartile for any base is less than 5 or if the median for any base is less than 20.
(PDF)

## Author Contributions

**Conceptualization:** Angelo Robotto, Carlotta Olivero, Elisa Pozzi, Claudia Strumia, Andrea Civra, Enrico Brizio, Denis Polato.

**Data curation:** Carlotta Olivero, Elisa Pozzi.

**Formal analysis:** Carlotta Olivero, Elisa Pozzi, Camilla Crasà, Cristina Fedele.

**Funding acquisition:** David Lembo.

**Investigation:** Carlotta Olivero, Elisa Pozzi, Camilla Crasà, Cristina Fedele, Maddalena Derosa, Massimo Di Martino, Stefania Latino, Giada Scorza.

**Methodology:** Carlotta Olivero, Elisa Pozzi.

**Project administration:** Angelo Robotto, Paola Quaglino, Denis Polato.

**Resources:** Angelo Robotto, Denis Polato.

**Validation:** Carlotta Olivero, Elisa Pozzi.

**Visualization:** Carlotta Olivero, Elisa Pozzi.

**Writing – original draft:** Carlotta Olivero, Elisa Pozzi, Claudia Strumia.

**Writing – review & editing:** Carlotta Olivero, Elisa Pozzi, Claudia Strumia, Andrea Civra, David Lembo, Paola Quaglino, Enrico Brizio, Denis Polato.

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
