## [Decision Letter · Decision Letter 0]

31 Jul 2023

PONE-D-23-20026Efficient wastewater sample filtration improves the detection of SARS-CoV-2 variants: an extensive analysis based on sequencing parametersPLOS ONE

Dear Dr. Olivero,

Thank you for submitting your manuscript to PLOS ONE. After careful consideration, we feel that it has merit but does not fully meet PLOS ONE’s publication criteria as it currently stands. Therefore, we invite you to submit a revised version of the manuscript that addresses the points raised during the review process.The authors are required to address the major and minor issues outlined by both Reviewer 1 and Reviewer 2. It is recommended that the authors consult a copyeditor to ensure that all typographical and grammatical have been addressed prior to resubmitting the manuscript. The statistical analysis needs to be revised and clearly described in the methods section. The results of the statistical analysis needs to be stated under the results section.

We look forward to receiving your revised manuscript.

Kind regards,

Jennifer Giandhari, Ph.D

Academic Editor

PLOS ONE

“Arpa Piemonte received no specific funding for this work.”

“The authors acknowledge the financial support received from Assicurazioni Generali and Intesa San Paolo to realize the Regional Centre of Molecular Biology run by Arpa Piemonte (Environmental Protection Agency of Piedmont) in La Loggia, Torino, Italy. This research was supported by EU funding within the MUR PNRR Extended Partnership initiative on Emerging Infectious Diseases (Project no. PE00000007, INF-ACT) to DL.”

“Arpa Piemonte received no specific funding for this work.”

5. We notice that your S2Fig is uploaded with the file type 'Figure'. Please amend the file type to 'Supporting Information'. Please ensure that each Supporting Information file has a legend listed in the manuscript after the references list.

Reviewers' comments:

Reviewer's Responses to Questions

**Comments to the Author**

1. Is the manuscript technically sound, and do the data support the conclusions?

Reviewer #1: Partly

Reviewer #2: Yes

2. Has the statistical analysis been performed appropriately and rigorously? 

Reviewer #1: No

Reviewer #2: No

3. Have the authors made all data underlying the findings in their manuscript fully available?

Reviewer #1: Yes

Reviewer #2: Yes

4. Is the manuscript presented in an intelligible fashion and written in standard English?

Reviewer #1: Yes

Reviewer #2: No

5. Review Comments to the Author

Reviewer #1: The authors focused on the use of wastewater surveillance to not only monitor the level of infection within a population but also to use for the early detection of variants. This paper is a response to recommendations from the European Union Commission to establish an approach to establish WW surveillance of SARS-CoV-2 and its variants. The authors aimed to obtain high-quality sequencing data to allow variants detection and compared for sample conditions and evaluated the best condition by assessing several sequencing parameters. The authors provided an overview of sequencing parameters to consider when the final goal is to detect variants.

It is important to use WW to monitor the level of infection within a population, but WW is also needed for the early detection of variants as the clinical sequencing data will continue to focus on symptomatic patients. Surveillance, quantification, sequencing, and comparison of results is intricate as wastewater is a complex matrix. Therefore, this study is valuable and important. However, the statistical analysis of the results needs to be revised and explained further before the proposed conclusions can be validated. Additionally, the high viral load found in the samples studied make it difficult to extrapolate the proposed results to a more “normal” scenario. The following issues need to be addressed before publication:

Issues

1- Lines 240-245. To compare the read frequencies differences between the four experimental conditions for three biological samples, the authors use the student’s t-test. There are requirements for applying the t-test: 1- normal distribution of the data obtained however, the sample number is too low to predict normal distribution.2- is used for pairwise comparisons, however four conditions are being compared. ANOVA for multiple comparisons is more suitable. After using a proper statistic test the significance of the differences detected should be reassessed.

2- The authors should follow the MIQE guidelines for reporting their qPCR results. The standard curve, ranging from 102 copies (cp)/μL to 0.5 cp/μL that is generated by serial 10-fold dilution does not provide enough points.

3- The authors do not provide an explanation of why the DNAse treatment and concentration of RNA in filtered samples increases the % of reads corresponding to Bacteria.

Minor issues

1- Line 129 describes 2 μL of TNA or purified RNA from each extracted sample was used for the one-step RT-qPCR. However, line 136 mentions 5μL of TNA/RNA for each WW sample and standard point.

2- Line 170 mentions ANOVA Analysis of variance but the results mention t-test significance.

Reviewer #2: The authors have conducted a study comparing methods for preparation of wastewater samples for amplicon sequencing of SARS-CoV-2, in particular assessing how the quality of sequencing data is impacted by prefiltering wastewater and implementing a step for purifying and concentrating extracted RNA. The authors conclude that filtration of wastewater samples leads to improved specificity and coverage of SARS-CoV-2 sequencing data, while treatment of RNA samples reduced the percentage of sequencing reads aligned to SARS-CoV-2 but had little impact on other sequencing metrics.

Although the experimental design and metrics used to assess SARS-CoV-2 sequencing quality were valid, the statistical methods used in analysis require improvement and clarification to draw valid conclusions. In particular, the statistical tests need to take into account the fact that samples exposed to the different treatments were derived from the same three wastewater samples and should therefore be treated as matched samples. Authors should also carry out comprehensive statistical tests (e.g. two-way ANOVA) to compare all the treatments together before implementing pairwise comparisons as they have currently done. Furthermore, the manuscript contains multiple linguistics errors and typos which need addressing, possibly through the use of a copyeditor.

Considering this, I recommend that this manuscript is publishable with major revisions.

Major issues

- The statistical methods used in data analysis require revision and clarification. Currently, the authors have stated in the methods (line 169-170) that they used “Student's t-test and ANOVA Analysis of variance” for statistical analysis. However, there is no mention of the results of ANOVA in the manuscript associated with any of the figures, making it unclear whether these tests were carried out. Furthermore, as mentioned above, the replicates within each treatment were derived from the same three wastewater samples, meaning statistical tests should consider that these replicates are matched between treatments. Finally, the two treatment variables that the authors are trying to determine the effects of (filtration and RNA treatment) are applied consecutively to these samples and are therefore nested, meaning these shouldn’t be considered as four separate treatments of one independent variable. The impact of these variables should be assessed by two-way ANOVA (or non-parametric equivalent, depending if parametric assumptions are met) to determine the overall effect of the two variables and any interaction between them.

- The manuscript contains grammatical, linguistic and typographical errors, which require revision. I have detailed a few of these in the minor issues. I recommend that the authors could consult a copyeditor to resolve this.

Minor Issues

- Line 33 – it is stated that previous studies are “ignoring” wastewater filtering and RNA purification, despite these factors being addressed in multiple studies (as referenced). Consider revising this statement.

- Line 35 – more information should be given on the treatments i.e i) filtering wastewater and i) DNAse treatment, purification and concentration of RNA.

- Line 54 – please define “WW” again in the main text.

- Line 54 – revise grammar in “was proposed as an epidemiologic tool approach…”

- Line 59 – “transmission among asymptomatic has also” needs amending e.g. asymptomatic hosts.

- Line 78-79 – The statement “amplicon sequencing for viral WGS is a preferable approach to library preparation” requires revision, as amplicon sequencing involves library preparation. Please revise this to state which method of library preparation you are comparing amplicon sequencing to.

- Line 82-83 – Please revise the grammar in “but not many monitoring have been”

- Authors names need to be added where references are used as part of sentences in the text, or sentence structure amended accordingly. This occurs in multiple locations in the manuscript e.g. Line 88 “…studies conducted by 21-22 showed…”.

- Line 97 – amend typo “ae”.

- Line 116 – Please clarify whether the centrifugation step mentioned in this line is an additional step to the centrifugation described in line 107, or a repeated description of the same step? If there was only one centrifugation, please remove from line 116 for clarity.

- Line 142 – please give final concentration of RP primer or include stock concentration used for dilution.

- Line 145 – please remove repetition of “reaction”.

- Line 153 – please state the unit of concentration (ng/ul?).

- Line 165 – please clarify whether one million PE reads was the aim for sequencing depth or whether this normalisation was achieved during bioinformatic analysis.

- Line 177 – please detail which programs were used for trimming and consensus genome generation.

- Line 180 – Please clarify what “minimum frequency” refers to – possibly minimum frequency of variations?

- Lines 194-199 – section “as detailed…followed by purification and concentration” not required as this is already described in the methods section.

- Figure 1 and Table S2 – please indicate whether there were any significant differences between treatments in these metrics.

- Line 235-236 – please clarify what is meant by “in intra- and inter-conditions”.

- Figures 1/3/4/5/6 – These plots are referred to as histograms, when they are bar plots with categorical variables on the x-axis.

- Figure 1 – Consider replacing this with a plot representing the data in Table S2, as this is what makes up the majority of results discussed in this section.

- Figure 2 – The colours in this figure are not clear, probably due to the low resolution of the image which also makes the axis labels illegible.

- Line 259 – Please clarify what is referred to as “lower lines”.

- Line 263 – It is unclear what is meant by “genomic contamination” and how this would impair sequencing – please discuss further to clarify.

- Line 283 – it is unclear whether all samples had a percentage of reads classified as SARS-CoV-2 of less than 40%, or just the means for these treatments are <40%.

- Line 289 – Please explain further the speculation that RNA sample concentration would lead to increased bacterial reads.

- Line 296-297 – Could you provide some discussion as to why wastewater filtration may improve quality scores?

- Section 3.4.2.2 – The percentage of reads assigned to SARS-CoV-2 was much higher with CLC workbench than Kraken. It would be good to comment on why this is, possibly due to raw reads used in Kraken classification including many short reads which are filtered out before direct mapping.

- Figure S2 is redundant as the % reads not mapped can be extrapolated from % mapped reads.

- Line 343 – The difference in % mapped reads is only significant between F-NT and F-T, but not between NF-NT and NF-T

- Line 347 – amend “off-target reads” as these are not reads at this point e.g off-target nucleic acid fragments.

- Line 346-347 – This finding is contrary to Mondal et al 2021 who found RNA purification and DNase treatment to increase % reads aligned to SARS-CoV-2, and Child et al 2023 (https://doi.org/10.1371/journal.pone.0284211) who found RNA purification and concentration to also increase alignment rate. Please discuss your findings in relation to these studies.

- Line 375 – amend “essential” as low frequency variants are still identified in unfiltered samples.

- Line 388-9 – Could you provide statistical tests to back up the statement that “mutation frequency was higher in the F conditions”? Also, please discuss why the mutation frequency would be higher in filtered samples if these mutations are associated with the BA.1 variant, which is stated to be dominant in the samples. Could the unfiltered treatments be picking up on more diversity of variants than filtered treatments, reducing the frequency of these typical BA.1 mutations in favour of other variants?

- Line 402 – Please amend this statement, as only the F-NT treatment passed for all samples, not F-T.

- Figure 8 – The colour scale used in this figure requires clarification. These percentages are clearly not directly related to the percentage parameters in the “Sequencing parameters” column. Is this scale relative to the highest and lowest value for each parameter amongst the treatments?

6. PLOS authors have the option to publish the peer review history of their article (what does this mean?). If published, this will include your full peer review and any attached files.

Reviewer #1: No

Reviewer #2: **Yes: **Harry T. Child

---

## [Author Response · Author response to Decision Letter 0]

23 Jan 2024

Dear Dr. Giandhari,

Thank you for your letter and the opportunity to submit the revised manuscript. We have addressed all the points raised in the editor’s and reviewers’ comments (bold sentences). Each point has been addressed, with changes tracked in the accompanying manuscript. We appreciate the chance to revise and submit our manuscript.

We revised the style requirements.

“Arpa Piemonte received no specific funding for this work.”

We thank the editor for these comments. We have amended and included the Financial Disclosure with these sentences: “Arpa Piemonte received no specific funding for this work. Professor DL was supported by EU funding within the MUR PNRR Extended Partnership initiative on Emerging Infectious Diseases (Project no. PE00000007, INF-ACT). The funders had no role in study design, data collection and analysis, decision to publish, or preparation of the manuscript.”.

“The authors acknowledge the financial support received from Assicurazioni Generali and Intesa San Paolo to realize the Regional Centre of Molecular Biology run by Arpa Piemonte (Environmental Protection Agency of Piedmont) in La Loggia, Torino, Italy. This research was supported by EU funding within the MUR PNRR Extended Partnership initiative on Emerging Infectious Diseases (Project no. PE00000007, INF-ACT) to DL.”

“Arpa Piemonte received no specific funding for this work.”

We have amended the Acknowledgement section: “Arpa Piemonte acknowledges the financial support received from Assicurazioni Generali and Intesa San Paolo to realize the Regional Centre of Molecular Biology in La Loggia, Torino, Italy.”

We commit to providing the accession numbers upon acceptance of the manuscript for publication.

5. We notice that your S2Fig is uploaded with the file type 'Figure'. Please amend the file type to 'Supporting Information'. Please ensure that each Supporting Information file has a legend listed in the manuscript after the references list.

We have removed S2 Fig as suggested by reviewer #2

Reviewer #1: The authors focused on the use of wastewater surveillance to not only monitor the level of infection within a population but also to use for the early detection of variants. This paper is a response to recommendations from the European Union Commission to establish an approach to establish WW surveillance of SARS-CoV-2 and its variants. The authors aimed to obtain high-quality sequencing data to allow variants detection and compared for sample conditions and evaluated the best condition by assessing several sequencing parameters. The authors provided an overview of sequencing parameters to consider when the final goal is to detect variants.

It is important to use WW to monitor the level of infection within a population, but WW is also needed for the early detection of variants as the clinical sequencing data will continue to focus on symptomatic patients. Surveillance, quantification, sequencing, and comparison of results is intricate as wastewater is a complex matrix. Therefore, this study is valuable and important. However, the statistical analysis of the results needs to be revised and explained further before the proposed conclusions can be validated. Additionally, the high viral load found in the samples studied make it difficult to extrapolate the proposed results to a more “normal” scenario. The following issues need to be addressed before publication:

Issues

1- Lines 240-245. To compare the read frequencies differences between the four experimental conditions for three biological samples, the authors use the student’s t-test. There are requirements for applying the t-test: 1- normal distribution of the data obtained however, the sample number is too low to predict normal distribution.2- is used for pairwise comparisons, however four conditions are being compared. ANOVA for multiple comparisons is more suitable. After using a proper statistic test the significance of the differences detected should be reassessed.

We thank the reviewer for these appropriate comments. We performed two-way ANOVA statistical tests for all the experiments but omitted to include the results in the manuscript. We have now stated ANOVA results for every figure of the manuscript. We revised statistical analysis throughout the manuscript. To assess the assumption of normal distribution of the data, we perform visual inspection analyses (histogram and Quantile-Quantile (QQ) plot) and statistical analysis (Shapiro-Wilk test) for each experiment. 

The assumption of normal distribution was statistically significant for each figure in the manuscript, so two-way ANOVA and t-test were applied. 

For lines 240-245 (now 253-259) we compared the copies input number for the different conditions. ANOVA and t-test results have been included in the manuscript as follows: 

- line 257: Two-way ANOVA (* = p ≤ 0.05), has been added.

- in line 172 paragraph 2.7 “statistical analysis” has been edited: “The normal distribution of the data was assessed by visual inspection analyses (histogram and Quantile-Quantile (QQ) plot) and statistical analysis (Shapiro-Wilk test). The significance of the data was calculated using two-way ANOVA (Analysis of variance) and paired Student's t-test.” 

2- The authors should follow the MIQE guidelines for reporting their qPCR results. The standard curve, ranging from 102 copies (cp)/μL to 0.5 cp/μL that is generated by serial 10-fold dilution does not provide enough points.

We made a mistake in reporting the data relative to the standard curve. The highest point of the standard curve is 103. We corrected the value in line 134

3- The authors do not provide an explanation of why the DNAse treatment and concentration of RNA in filtered samples increases the % of reads corresponding to Bacteria.

We thank the reviewer for making this point. 

After reviewing the statistics, we have removed the sentence about the concentration step being the reason for the increase of bacteria reads %.

Minor issues

1- Line 129 describes 2 μL of TNA or purified RNA from each extracted sample was used for the one-step RT-qPCR. However, line 136 mentions 5μL of TNA/RNA for each WW sample and standard point. 

We have made a mistake. The correct volume is 5 μL (line 138). 

2- Line 170 mentions ANOVA Analysis of variance but the results mention t-test significance.

We have stated the ANOVA results in the caption of each figure in the manuscript. 

Reviewer #2: The authors have conducted a study comparing methods for preparation of wastewater samples for amplicon sequencing of SARS-CoV-2, in particular assessing how the quality of sequencing data is impacted by prefiltering wastewater and implementing a step for purifying and concentrating extracted RNA. The authors conclude that filtration of wastewater samples leads to improved specificity and coverage of SARS-CoV-2 sequencing data, while treatment of RNA samples reduced the percentage of sequencing reads aligned to SARS-CoV-2 but had little impact on other sequencing metrics.

Although the experimental design and metrics used to assess SARS-CoV-2 sequencing quality were valid, the statistical methods used in analysis require improvement and clarification to draw valid conclusions. In particular, the statistical tests need to take into account the fact that samples exposed to the different treatments were derived from the same three wastewater samples and should therefore be treated as matched samples. Authors should also carry out comprehensive statistical tests (e.g. two-way ANOVA) to compare all the treatments together before implementing pairwise comparisons as they have currently done. Furthermore, the manuscript contains multiple linguistics errors and typos which need addressing, possibly through the use of a copyeditor.

Considering this, I recommend that this manuscript is publishable with major revisions.

Major issues

- The statistical methods used in data analysis require revision and clarification. Currently, the authors have stated in the methods (line 169-170) that they used “Student's t-test and ANOVA Analysis of variance” for statistical analysis. However, there is no mention of the results of ANOVA in the manuscript associated with any of the figures, making it unclear whether these tests were carried out. Furthermore, as mentioned above, the replicates within each treatment were derived from the same three wastewater samples, meaning statistical tests should consider that these replicates are matched between treatments. Finally, the two treatment variables that the authors are trying to determine the effects of (filtration and RNA treatment) are applied consecutively to these samples and are therefore nested, meaning these shouldn’t be considered as four separate treatments of one independent variable. The impact of these variables should be assessed by two-way ANOVA (or non-parametric equivalent, depending if parametric assumptions are met) to determine the overall effect of the two variables and any interaction between them.

We thank the reviewer for these appropriate comments. We have revised the statistical analysis throughout the manuscript. We have considered the samples as matched, as the same sample is exposed to the different treatments, as the reviewer points out. We performed two-way ANOVA statistical tests for all the experiments but omitted to include the results in the manuscript. We have now reported the ANOVA results in every caption of the manuscript. 

- in lines 172-175: Paragraph 2.7 “statistical analysis” was amended to read: “The normal distribution of the data was assessed by visual inspection analyses (histogram and Quantile-Quantile (QQ) plot) and statistical analysis (Shapiro-Wilk test). The significance of the data was calculated using two-way ANOVA (Analysis of variance) and paired Student's t-test.” 

- The manuscript contains grammatical, linguistic and typographical errors, which require revision. I have detailed a few of these in the minor issues. I recommend that the authors could consult a copyeditor to resolve this.

We thank the reviewer for pointing out the typographical errors. The manuscript has been grammatically revised. 

Minor Issues

- Line 33 – it is stated that previous studies are “ignoring” wastewater filtering and RNA purification, despite these factors being addressed in multiple studies (as referenced). Consider revising this statement.

We agree with the reviewer’s comment. We amend line 33. The sentence reads as follows: “Therefore, different groups have approached the issue with different strategies, mainly focusing on WW concentration methods, but only a few groups highlight the importance of prefiltering WW samples and/or purification of RNA samples.” 

- Line 35 – more information should be given on the treatments i.e i) filtering wastewater and i) DNAse treatment, purification and concentration of RNA.

We thank the reviewer for asking us to clarify this sentence. We amended the sentence from line 34 to read: “Aiming to obtain high-quality sequencing data allowing variants detection, we compared four sample conditions generated from the treatment of: i) WW samples by WW filtration and ii) the extracted RNA by DNAse treatment, purification and concentration of the extracted RNA.” 

- Line 54 – please define “WW” again in the main text. 

We have defined WW abbreviation in line 55. 

- Line 54 – revise grammar in “was proposed as an epidemiologic tool approach…”

In lines 53-55 the sentence has been revised with: “In the midst of the pandemic caused by Severe Acute Respiratory Syndrome Coronavirus - 2 (SARS-CoV-2), Wastewater Based Epidemiology (WBE) was proposed as an epidemiological tool approach for wastewater (WW) surveillance. 

- Line 59 – “transmission among asymptomatic has also” needs amending e.g. asymptomatic hosts.

We added the word “hosts” in line 60. 

- Line 78-79 – The statement “amplicon sequencing for viral WGS is a preferable approach to library preparation” requires revision, as amplicon sequencing involves library preparation. Please revise this to state which method of library preparation you are comparing amplicon sequencing to.

We revised the sentence: “Whole-genome sequencing (WGS) of viruses has become a powerful tool for studying emerging infectious diseases while monitoring public health and amplicon sequencing for viral WGS is the preferable approach for library preparation since it is simple, cost-effective and sensitive for detecting genomes from samples with low viral loads.” in lines 78-81. 

- Line 82-83 – Please revise the grammar in “but not many monitoring have been”

We revised the grammar in lines 84. 

- Authors names need to be added where references are used as part of sentences in the text, or sentence structure amended accordingly. This occurs in multiple locations in the manuscript e.g. Line 88 “…studies conducted by 21-22 showed…”.

We amended the sentences according to this comment. 

- Line 97 – amend typo “ae”. 

We amended the typo. 

- Line 116 – Please clarify whether the centrifugation step mentioned in this line is an additional step to the centrifugation described in line 107, or a repeated description of the same step? If there was only one centrifugation, please remove from line 116 for clarity. 

We appreciate the reviewer's request for clarification on this point.

The centrifugation step in line 109 refers to the samples that undergo filtration. 

The centrifugation step in line 118 refers to the non-filtered samples.

We amended line 109 to read: “Before the filtration step, 40mL of the composite WW samples selected for filtration were centrifuged at 3000 x g for 10 minutes to remove solids.” 

- Line 142 – please give final concentration of RP primer or include stock 

---

## [Decision Letter · Decision Letter 1]

4 Apr 2024

PONE-D-23-20026R1Efficient wastewater sample filtration improves the detection of SARS-CoV-2 variants: an extensive analysis based on sequencing parametersPLOS ONE

Dear Dr. Olivero,

Thank you for submitting your manuscript to PLOS ONE. After careful consideration, we feel that it has merit but does not fully meet PLOS ONE’s publication criteria as it currently stands. Therefore, we invite you to submit a revised version of the manuscript that addresses the points raised during the review process.

Please complete the minor revisions as given below, before the manuscript is fully accepted for publication. 

We look forward to receiving your revised manuscript.

Kind regards,

Poovendhree Reddy

Academic Editor

PLOS ONE

Journal Requirements:

Additional Editor Comments:

Please ensure that the quality of your figures is improved in the next revision. 

Reviewers' comments:

Reviewer's Responses to Questions

**Comments to the Author**

1. If the authors have adequately addressed your comments raised in a previous round of review and you feel that this manuscript is now acceptable for publication, you may indicate that here to bypass the “Comments to the Author” section, enter your conflict of interest statement in the “Confidential to Editor” section, and submit your "Accept" recommendation.

Reviewer #2: (No Response)

2. Is the manuscript technically sound, and do the data support the conclusions?

Reviewer #2: Yes

3. Has the statistical analysis been performed appropriately and rigorously? 

Reviewer #2: Yes

4. Have the authors made all data underlying the findings in their manuscript fully available?

Reviewer #2: Yes

5. Is the manuscript presented in an intelligible fashion and written in standard English?

Reviewer #2: Yes

6. Review Comments to the Author

Reviewer #2: The authors have conducted a study comparing methods for preparation of wastewater samples for amplicon sequencing of SARS-CoV-2, in particular assessing how the quality of sequencing data is impacted by prefiltering wastewater and implementing a step for purifying and concentrating extracted RNA. The authors conclude that filtration of wastewater samples leads to improved specificity and coverage of SARS-CoV-2 sequencing data, while treatment of RNA samples reduced the percentage of sequencing reads aligned to SARS-CoV-2 but had little impact on other sequencing metrics.

The experimental design and metrics used to assess SARS-CoV-2 sequencing quality are valid, and following amendments in response to the initial round of reviewers comments the statistical methods used are now valid and clear. Considering this, I recommend that this manuscript requires only minor revisions. I understand that these are largely grammatical suggestions, but I hope the authors appreciate the improvement in clarity that they can provide.

Minor Issues

- Figure legends – Two way ANOVA results are currently givenm in the format “Two-way ANOVA (* = p ≤ 0.05).”. Please amend these without the asterisk as otherwise it seems to be referring to the asterisks in the plot which are the paired T test results e.g. “Two-way ANOVA (p ≤ 0.05)”.

- Figure 2 – The resolution of Figure 2 is still very low in the version I have been sent. This should be checked before publication.

- Line 54 – Suggested revision of “Wastewater Based Epidemiology (WBE) was proposed as an epidemiological tool approach for wastewater (WW) surveillance” to “Wastewater Based Epidemiology (WBE) was proposed as an epidemiological tool approach for disease surveillance”

- Line 59 – “asymptomatic host” needs amending to “asymptomatic hosts”

- Line 80-83 – Suggested revision of “However, most of the viral genome sequences used to monitor viruses’ evolution have been“ to “However, most of the viral genome sequences used to monitor the evolution of viruses have been“, and “but not many monitoring have been performed” to “but not much monitoring has been performed”

- Line 87-89 – please amend to “Earlier than the development of WBE for the COVID-19 pandemic, previous studies showed that the detection of viral pathogens in sewage depended on the concentration and extraction methods used (21–23).”

- Line 248-249 – may I suggest an amendment to “Moreover, the number of viral cp input into library preparation was not identical within or between conditions...”

7. PLOS authors have the option to publish the peer review history of their article (what does this mean?). If published, this will include your full peer review and any attached files.

Reviewer #2: **Yes: **Dr Harry T. Child

---

## [Author Response · Author response to Decision Letter 1]

8 Apr 2024

Additional Editor Comments:

Please ensure that the quality of your figures is improved in the next revision. 

The quality of the figures has been checked and improved through the PACE digital diagnostic tool.

Reviewers' comments:

Reviewer #2: The authors have conducted a study comparing methods for preparation of wastewater samples for amplicon sequencing of SARS-CoV-2, in particular assessing how the quality of sequencing data is impacted by prefiltering wastewater and implementing a step for purifying and concentrating extracted RNA. The authors conclude that filtration of wastewater samples leads to improved specificity and coverage of SARS-CoV-2 sequencing data, while treatment of RNA samples reduced the percentage of sequencing reads aligned to SARS-CoV-2 but had little impact on other sequencing metrics.

The experimental design and metrics used to assess SARS-CoV-2 sequencing quality are valid, and following amendments in response to the initial round of reviewers comments the statistical methods used are now valid and clear. Considering this, I recommend that this manuscript requires only minor revisions. I understand that these are largely grammatical suggestions, but I hope the authors appreciate the improvement in clarity that they can provide.

Minor Issues

- Figure legends – Two way ANOVA results are currently givenm in the format “Two-way ANOVA (* = p ≤ 0.05).”. Please amend these without the asterisk as otherwise it seems to be referring to the asterisks in the plot which are the paired T test results e.g. “Two-way ANOVA (p ≤ 0.05)”.

We thank the reviewer for this appropriate comment. It helps to improve the clarity of the result. We have amended all the sentences in which the ANOVA result was stated.

- Figure 2 – The resolution of Figure 2 is still very low in the version I have been sent. This should be checked before publication.

The resolution of Figure 2 has been improved. 

- Line 54 – Suggested revision of “Wastewater Based Epidemiology (WBE) was proposed as an epidemiological tool approach for wastewater (WW) surveillance” to “Wastewater Based Epidemiology (WBE) was proposed as an epidemiological tool approach for disease surveillance”

We have amended the sentence. We have added the word “wastewater” before “WW” in line 57.

- Line 59 – “asymptomatic host” needs amending to “asymptomatic hosts”

We have amended the word.

- Line 80-83 – Suggested revision of “However, most of the viral genome sequences used to monitor viruses’ evolution have been“ to “However, most of the viral genome sequences used to monitor the evolution of viruses have been“, and “but not many monitoring have been performed” to “but not much monitoring has been performed”

We have amended the sentences.

- Line 87-89 – please amend to “Earlier than the development of WBE for the COVID-19 pandemic, previous studies showed that the detection of viral pathogens in sewage depended on the concentration and extraction methods used (21–23).”

We have amended the sentence.

- Line 248-249 – may I suggest an amendment to “Moreover, the number of viral cp input into library preparation was not identical within or between conditions...”

We have amended the sentence.

---

## [Decision Letter · Decision Letter 2]

8 May 2024

Efficient wastewater sample filtration improves the detection of SARS-CoV-2 variants: an extensive analysis based on sequencing parameters

PONE-D-23-20026R2

Dear Dr. Olivero,

We’re pleased to inform you that your manuscript has been judged scientifically suitable for publication and will be formally accepted for publication once it meets all outstanding technical requirements.

Kind regards,

Ricardo Santos

Academic Editor

PLOS ONE

Additional Editor Comments (optional):

Reviewers' comments:

Reviewer's Responses to Questions

**Comments to the Author**

1. If the authors have adequately addressed your comments raised in a previous round of review and you feel that this manuscript is now acceptable for publication, you may indicate that here to bypass the “Comments to the Author” section, enter your conflict of interest statement in the “Confidential to Editor” section, and submit your "Accept" recommendation.

Reviewer #2: All comments have been addressed

2. Is the manuscript technically sound, and do the data support the conclusions?

Reviewer #2: Yes

3. Has the statistical analysis been performed appropriately and rigorously? 

Reviewer #2: Yes

4. Have the authors made all data underlying the findings in their manuscript fully available?

Reviewer #2: Yes

5. Is the manuscript presented in an intelligible fashion and written in standard English?

Reviewer #2: Yes

6. Review Comments to the Author

Reviewer #2: (No Response)

7. PLOS authors have the option to publish the peer review history of their article (what does this mean?). If published, this will include your full peer review and any attached files.

Reviewer #2: **Yes: **Harry T. Child

---

## [Editor Report · Acceptance letter]

14 May 2024

PONE-D-23-20026R2 

PLOS ONE

Dear Dr. Olivero, 

I'm pleased to inform you that your manuscript has been deemed suitable for publication in PLOS ONE. Congratulations! Your manuscript is now being handed over to our production team.

Kind regards, 

on behalf of

Dr. Ricardo Santos 

Academic Editor

PLOS ONE